# Commander-GPT: Dividing and Routing for Multimodal Sarcasm Detection

**Yazhou Zhang**  *yzhou_zhang@tju.edu.cn*
*School of Computer Science and Technology, Tianjin University*

**Chunwang Zou**  *1376132667@qq.com*
*Software Engineering College, Zhengzhou University of Light Industry*

**Bo Wang**  *bo_wang@tju.edu.cn*
*School of Computer Science and Technology, Tianjin University*

**Jing Qin**[*]  *harry.qin@polyu.edu.hk*
*Center for Smart Health, School of Nursing, The Hong Kong Polytechnic University*

**Prayag Tiwari**[*]  *prayag.tiwari@ieee.org*
*School of Information Technology, Halmstad University*

**Reviewed on OpenReview:** *https://openreview.net/forum?id=zRxRbBsqwE*

## Abstract

Multimodal sarcasm understanding is a high-order cognitive task. Although large language models (LLMs) have shown impressive performance on many downstream NLP tasks, growing evidence suggests that they struggle with sarcasm understanding. In this paper, we propose Commander-GPT, a modular decision routing framework inspired by military command theory. Rather than relying on a single LLM's capability, Commander-GPT orchestrates a team of specialized LLM agents where each agent will be selectively assigned to a focused sub-task such as context modeling, sentiment analysis, etc. Their outputs are then routed back to the commander, which integrates the information and performs the final sarcasm judgment. To coordinate these agents, we introduce three types of centralized commanders: (1) a trained lightweight encoder-based commander (e.g., multi-modal BERT); (2) four small autoregressive language models, serving as moderately capable commanders (e.g., DeepSeek-VL); (3) two large LLM-based commander (Gemini Pro and GPT-4o) that performs task routing, output aggregation, and sarcasm decision-making in a zero-shot fashion. We evaluate Commander-GPT on the MMSD and MMSD 2.0 benchmarks, comparing five prompting strategies. Experimental results show that our framework achieves4.4% and 8.5% improvement in F1 score over state-of-the-art (SoTA) baselines on average, demonstrating its effectiveness.

## 1 Introduction

The era of large language models (LLMs) has been propelled by the scaling laws of language models and the emergence of capabilities with increasing model scale. SoTA LLMs, such as GPT-4o (Achiam et al., 2023), Claude 4[1], DeepSeek R1 (DeepSeek-AI et al., 2025), Qwen 3 (Yang et al., 2025), et., have demonstrated

---

[*]Corresponding authors.
[1]https://www.anthropic.com/news/claude-4

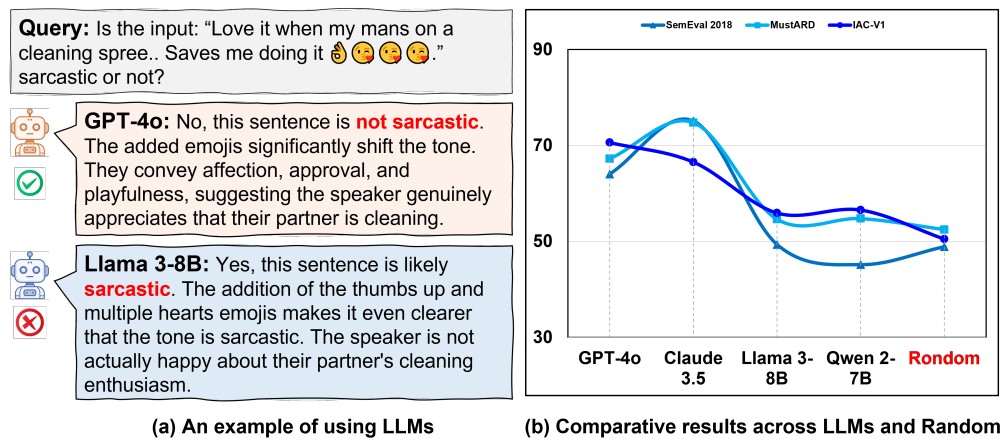

Figure 1: LLM Performance on three sarcasm datasets in prior work.

remarkable performance across a wide range of downstream natural language processing (NLP) tasks, including question answering, machine translation, commonsense reasoning, and code generation. These models exhibit impressive zero-shot and few-shot generalization abilities, leading to the belief that LLMs may have reached a critical threshold of general intelligence (Zhang et al., 2025).

Despite these advances, sarcasm understanding remains a significant and unresolved challenge. Sarcasm is a nuanced linguistic phenomenon that often employs rhetorical devices such as irony, hyperbole, and contradiction to express sentiments that diverge sharply from the literal meanings of words (Liu et al., 2023b). For example, the sentence "Oh great, another meeting that could have been an email." appears to express enthusiasm, but in reality, conveys frustration or annoyance, its sarcastic tone hinging on pragmatic context and emotional subtext. Accurate sarcasm detection requires a combination of contextual reasoning, emotional inference, and figurative language interpretation, multi-modal interaction, etc., all of which demand high-order cognitive capabilities.

The growing evidence proves that even state-of-the-art models such as GPT-4, Claude 3.5 often perform poorly on sarcasm detection, sometimes close to random guessing, as shown in Fig. 1 (Yao et al., 2025). This observation challenges the prevailing assumption that general-purpose LLMs, by virtue of their scale, can seamlessly handle high-level pragmatic tasks. This discrepancy reveals a fundamental limitation: current methods treat sarcasm detection as an undivided, end-to-end task, relying on a single model to implicitly execute multiple layers of reasoning. These include sentiment recognition, rhetorical structure identification, contextual interpretation, and visual-textual alignment. Without explicit modeling of these sub-tasks, even powerful LLMs often fail to capture the complex and composite nature of sarcasm, particularly in multimodal scenarios.

To this end, we propose Commander-GPT, a structured multi-agent framework that decomposes sarcasm detection into six cognitively meaningful sub-tasks. They are: *context modeling*, *sentiment analysis*, *rhetorical device recognition*, *facial expression recognition*, *image summarization*, and *scene text recognition*. Each sub-task is handled by an expert LLM or MLLM agent. Rather than invoking all agents for every input, the commander first analyzes the input and then activates only the agents that are most suitable for handling the relevant sub-tasks by introducing the routing scorer. In addition, BLIP-2 (Li et al., 2022), Vision Transformer[2], and OCR-2.0 (Wei et al., 2024) are selected as vision agents for image summarization, facial expression recognition, and scene text recognition, respectively, while Llama 3-8B[3], Qwen 2.5-1.5B[4], and RoBERTa[5] serve as linguistic specialists. Their outputs are then routed back to the commander, which integrates the information and performs the final sarcasm judgment.

---

[2] https://huggingface.co/motheecreator/vit-Facial-Expression-Recognition
[3] https://huggingface.co/meta-llama/Meta-Llama-3-8B
[4] https://huggingface.co/THUDM/Qwen2.5-1.5B
[5] https://huggingface.co/SamLowe/roberta-base-go_emotions

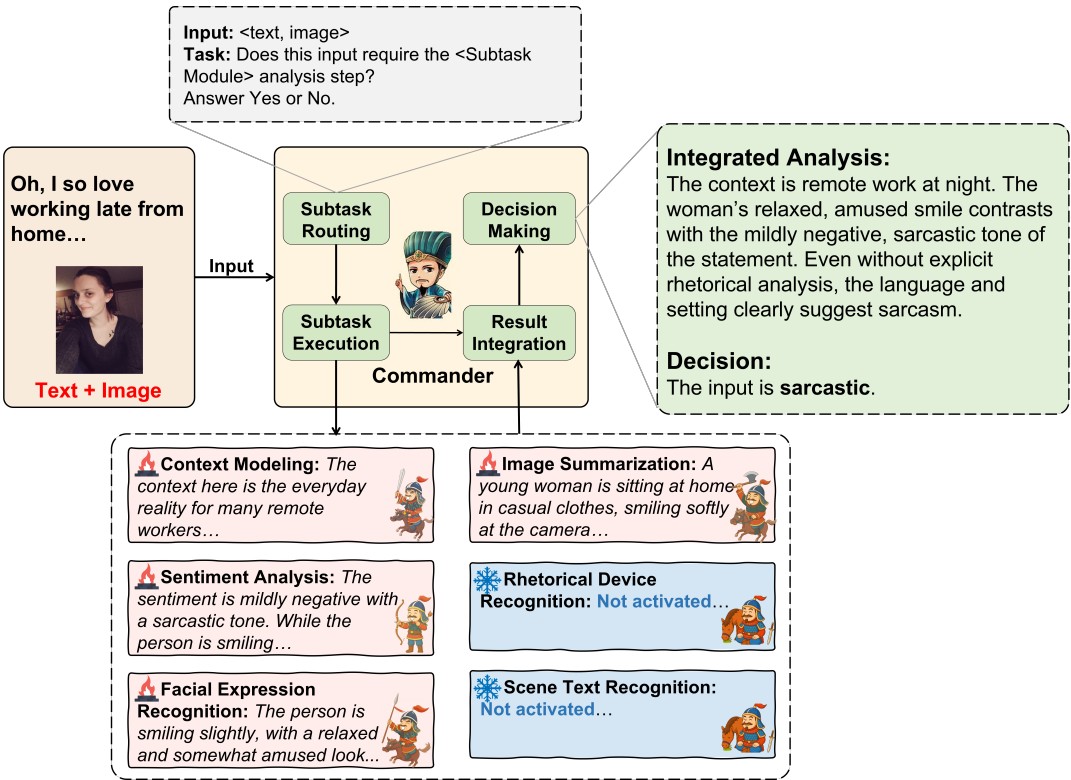

Figure 2: The overall architecture of Commander-GPT.

To coordinate these agents, we introduce three types of centralized commanders: (1) a trained lightweight encoder-based commander (e.g., multi-modal BERT); (2) four small autoregressive language models, serving as moderately capable commanders (e.g., DeepSeek-VL-7B); (3) two large LLM-based commander (Gemini Pro and GPT-4o) that performs task routing, output aggregation, and sarcasm decision-making. This diverse commander configuration enables a comprehensive exploration of the trade-offs between performance and scalability.

Finally, we conduct empirical evaluations of Commander-GPT on two benchmark datasets: MMSD and MMSD 2.0. We compare our framework against five state-of-the-art (SoTA) prompting strategies (e.g., Chain-of-Thought, Plan-and-Solve, $S^3$ Agent, etc.). Experimental results highlight three key observations: (1) Commander-GPT achieves 4.4% and 8.5% improvement in F1 score over strong baselines, demonstrating the effectiveness; (2) despite advances in large multimodal language models (MLLMs), fine-tuned small models (such as BERT) still exhibit stronger sarcasm detection capabilities; (3) the Commander-GPT model demonstrates robust generalization across diverse backbone LLMs and domains.

Our contributions are summarized as follows:

- We propose **Commander-GPT**, a modular multi-agent framework for multimodal sarcasm detection.

- We introduce a set of centralized commanders with varying model capacities and conduct a systematic comparison of their orchestration ability in sarcasm detection.

- We present extensive experiments on MMSD and MMSD 2.0, showing that Commander-GPT achieves 4.4% and 8.5% F1 improvement over SoTA baselines on average.

## 2 The Proposed Approach

### 2.1 Problem Formulation

Consider a multimodal sarcasm detection task where we are given a dataset $\mathcal{D} = (x_i, y_i)_{i=1}^{N}$, where each instance $x_i = (t_i, v_i)$ consists of a textual component $t_i \in \mathcal{T}$ and a visual component $v_i \in \mathcal{V}$, and $y_i \in 0, 1$ represents the binary sarcasm label. The objective is to learn a function $f : \mathcal{T} \times \mathcal{V} \to 0, 1$ that accurately predicts sarcasm labels.

Traditional approaches employ a single monolithic model $\mathcal{M}$ to directly map the multimodal input to the prediction: $f(t, v) = \mathcal{M}(t, v)$. However, as established in prior work, this paradigm suffers from the inherent complexity of sarcasm understanding, which requires simultaneous processing of multiple linguistic and visual cues that may exhibit semantic contradictions. In this work, we propose a fundamentally different approach inspired by military command structures. We decompose the complex sarcasm detection task into $K$ specialized subtasks, where each subtask $\tau_k$ for $k \in 1, 2, ..., K$ focuses on extracting specific types of information.

We assume access to a collection of expert models $\{\mathcal{A} = A_1, A_2, ..., A_M\}$, where each agent $A_j$ has varying capabilities across different subtasks. The capability of agent $A_j$ for subtask $\tau_k$ is characterized by a performance score $\pi_{j,k} \in [0, 1]$, where higher values indicate better suitability. The core challenge lies in learning an optimal assignment function $\{\alpha : 1, 2, ..., K \to 1, 2, ..., M\}$ that maps each subtask $\tau_k$ to the most appropriate agent $A_{\alpha(k)}$. This assignment should maximize the overall system performance while considering the complementary strengths of different agents.

Given the assignment function $\alpha$, each subtask $\tau_k$ produces an intermediate output $z_k = \phi_k(x_i)$ through the assigned agent $A_{\alpha(k)}$. The final prediction is made by a commander model $\mathcal{C}$ that aggregates all intermediate representations:

$$f(t_i, v_i) = \mathcal{C}(z_k{}_{k=1}^{K}) = \mathcal{C}\{\phi_1(x_i), \phi_2(x_i), \phi_3(x_i), ..., \phi_K(x_i)\} \tag{1}$$

where $x_i \in \{t_i, v_i\}$.

The commander model $\mathcal{C}$ serves as the central decision-making unit, analogous to a military command center that processes intelligence reports from specialized units and makes strategic decisions. This hierarchical architecture enables specialized processing while maintaining global coordination through the commander's integrative reasoning capabilities.

It is important to note that while the overall objective can be expressed as a joint minimization over the routing assignment $\alpha$ and the commander parameters $\theta_{\mathcal{C}}$, in practice we adopt a *staged training* procedure rather than full end-to-end optimization. Specifically, the routing classifier is first trained with GPT-4o–distilled supervision to learn $\alpha$, and then the commander is trained on top of the routed outputs. This process can be formulated as a bilevel optimization:

$$\alpha^{\star} = \arg\min_{\alpha} \mathbb{E}_{(x, T_k)}[\mathcal{L}_{\text{route}}(r_{ik}, p_k(x; \alpha))], \tag{2}$$

$$\theta_{\mathcal{C}}^{\star} = \arg\min_{\theta_{\mathcal{C}}} \mathbb{E}_{(x, y)}[\mathcal{L}_{\text{cmd}}(y, f(x; \alpha^{\star}, \theta_{\mathcal{C}}))], \tag{3}$$

where $\mathcal{L}_{\text{route}}$ denotes the binary cross-entropy loss for routing decisions and $\mathcal{L}_{\text{cmd}}$ denotes the binary sarcasm detection loss. In this work $\alpha$ is optimized first and held fixed when training the commander. Exploring fully differentiable end-to-end optimization of router and commander is left for future research.

### 2.2 Subtask Division

Sarcasm understanding often involves: (1) understanding the broader context and pragmatic implications, (2) recognizing the surface-level emotional expression, (3) detecting linguistic markers such as irony and hyperbole, (4) interpreting visual emotional cues that may contradict textual sentiment, (5) understanding the visual scene context, and (6) processing any textual information embedded in images, as shown in Fig. 2. Based on this theoretical foundation, we decompose the complex sarcasm detection task into $K = 6$ cognitively meaningful subtasks, where each subtask $\tau_k$ focuses on extracting specific types of information that contribute to sarcasm understanding:

- $\tau_1$: **Context Modeling**: Analyze broader conversational context and pragmatic implications from text $t$: $\phi_1(t) \to \mathcal{P}$, where $\mathcal{P}$ denotes the pragmatic interpretation space.

- $\tau_2$: **Sentiment Analysis**: Extract fine-grained emotional polarity from text $t$: $\phi_2(t) \to \mathcal{S}$, where $\mathcal{S} \in \mathbb{R}^{28}$ represents multi-dimensional emotion categories (e.g., joy, anger, surprise). This identifies the surface-level emotional expression.

- $\tau_3$: **Rhetorical Device Recognition**: Identify linguistic patterns and rhetorical structures in text $t$: $\phi_3(t) \to \mathcal{R}$, where $\mathcal{R}$ represents rhetorical markers such as irony, hyperbole, and understatement.

- $\tau_4$: **Facial Expression Recognition**: Extract facial emotions from image $v$: $\phi_4(v) \to \mathcal{E}$, where $\mathcal{E} \in \mathbb{R}^7$ represents basic emotion categories (happy, sad, angry, fear, surprise, disgust, neutral).

- $\tau_5$: **Image Summarization**: Generate comprehensive scene description from image $v$: $\phi_5(v) \to \mathcal{D}$, where $\mathcal{D}$ represents natural language descriptions of visual content and scene context.

- $\tau_6$: **Scene Text Recognition**: Extract textual content embedded in image $v$: $\phi_6(v) \to \mathcal{T}scene$, where $\mathcal{T}scene$ represents text sequences found in the visual scene.

This decomposition is motivated by the observation that sarcasm often emerges from contradictions between these different information layers. For instance, a text expressing positive sentiment ($\tau_2$) accompanied by a person's annoyed facial expression ($\tau_4$) in a chaotic scene ($\tau_5$) may indicate sarcastic intent. By explicitly modeling these components, our framework can capture the nuanced interplay that previous approaches often miss.

## 2.3 Routing Approach

The routing scorer is a core component in Commander-GPT, enabling the commander to dynamically select relevant sub-task agents for a given input $x = (t, v)$. We provide two representative instantiations: a learnable classifier (e.g., BERT-based commander) and a prompt-based approach for LLMs (e.g., GPT-4o, DeepSeek VL).

**Learnable Routing Classifier (Multimodal BERT based Commander).** Let $K$ denote the total number of sub-tasks $\{\tau_k\}_{k=1}^K$. For each input $x = (t, v)$, we construct $K$ paired samples $(x, T_k)$, where $T_k$ is the descriptor for sub-task $\tau_k$ (e.g., "sentiment analysis"). The routing scorer outputs an activation probability for each sub-task:

$$p_k(x) = P_\theta(r_k = 1 \mid x, T_k) = \sigma\left(\mathbf{w}_k^\top h_{\mathrm{CLS}}(x, T_k) + b_k\right) \tag{4}$$

where $h_{\mathrm{CLS}}(x, T_k) = \mathrm{Fusion}(\mathrm{BERT}(t), \mathrm{ViT}(v), T_k)$ denotes the fused multimodal representation obtained by combining the BERT-encoded text $t$, the ViT-encoded image $v$, and the sub-task descriptor $T_k$. Here, $\mathrm{Fusion}(\cdot)$ can be implemented as feature concatenation. The parameters $\mathbf{w}_k, b_k$ are trainable for each sub-task, and $\sigma$ is the sigmoid function.

To train the routing classifier, we require a labeled routing dataset $\mathcal{D}_{\mathrm{route}} = \{(x_i, T_k, r_{ik})\}$. We distill agent activation decisions from a powerful instruction-tuned vision-language model, GPT-4o. Specifically, for each training sample $x_i = (t_i, v_i)$ and each sub-task descriptor $T_k$, we prompt GPT-4o with a question such as:

> **Input**: <t_i, v_i>
> **Task**: Does this input require the "<T_k>" analysis step?
> Answer "Yes" or "No".

The binary response is recorded as the activation label $r_{ik} \in \{0, 1\}$. In this work, we distill approximately 5,000 routing supervision instances, enabling scalable and reliable training of the routing classifier.

The routing scorer is trained by minimizing the binary cross-entropy loss:

$$\mathcal{L}_{\mathrm{route}} = -\frac{1}{NK} \sum_{i=1}^N \sum_{k=1}^K \left[r_{ik} \log p_k(x_i) + (1 - r_{ik}) \log(1 - p_k(x_i))\right] \tag{5}$$

During inference, agent $A_k$ is activated for $x$ if $p_k(x) > \alpha_k$, where $\alpha_k \in [0, 1]$ is a tunable threshold.

**Prompt-based Routing (LLM Commander).** For instruction-tuned LLMs (e.g., GPT-4o, DeepSeek VL), agent routing is formulated as a natural language inference task. For each $(x, T_k)$, we construct the prompt (see App. A3).

The LLM $\mathcal{M}_{\text{LLM}}$ is queried with $\mathcal{I}(x, T_k)$ and returns a textual response, which is mapped to $r_k(x) \in \{0, 1\}$ (1 if "Yes", 0 if "No"). This approach requires no explicit training and leverages the zero-shot or few-shot abilities of modern LLMs.

**GPT-4o Distillation.** To obtain supervision for the routing classifier, we distilled activation decisions from GPT-4o on approximately 5000 $(x, T_k)$ pairs sampled from the training split across all six subtasks. Each query was designed as a zero-shot, template-based prompt with strict JSON-formatted output (please refer to App. A.5), using temperature $= 0.0$ and top-$p = 1.0$ to minimize randomness. If GPT-4o's response did not conform to the schema, we automatically re-issued the prompt up to two times; invalid cases ($< 1.2\%$) were discarded. Prompts were stratified to balance text-only, image-only, and multimodal inputs, as well as cases with and without facial expressions or scene text, ensuring broad coverage and preventing train–test leakage.

Each instance was queried three times independently, and inter-run agreement reached a Cohen's $\kappa$ of 0.82, indicating strong consistency. For sentiment, majority voting across the three runs was adopted; for routing, items with disagreement were marked low-confidence and down-weighted during training. In addition, two human annotators reviewed 200 randomly sampled cases, achieving inter-annotator $\kappa = 0.79$ and 87% agreement between GPT-4o and human consensus. These results suggest that the distilled labels are of sufficiently high quality and stability to provide reliable supervision for training the routing classifier.

## 2.4 Subtask Execution

Once the routing scorer identifies the relevant sub-task agents for a given input $x_i = (t_i, v_i)$, each activated agent $A_k \in \mathcal{A}_{x_i}$ independently processes its assigned sub-task $\tau_k$. These specialized agents operate in parallel, leveraging their unique capabilities to extract specific types of information from the textual and visual components of $x_i$.

**Linguistic Specialists.** For the textual component $t_i$ of $x_i$, three linguistic specialist agents are employed:

- **Context Modeling Agent (Llama 3-8B):** Analyzes broader conversational context and pragmatic implications via: $z_1 = \phi_1(t_i) = \text{Llama 3}(\text{"Analyze the contextual implications: "} + t_i)$, where $z_1$ captures deeper meaning and situational context.

- **Sentiment Analysis Agent (RoBERTa):** Extracts fine-grained sentiment polarity using a fine-tuned RoBERTa: $z_2 = \phi_2(t_i) = \text{RoBERTa}_{\text{emotions}}(t_i) \in \mathbb{R}^3$, a 3D vector representing *positive*, *neutral*, and *negative* sentiments.

- **Rhetorical Device Agent (Qwen 2.5-1.5B):** Identifies linguistic patterns and rhetorical structures (irony, hyperbole, understatement) using: $z_3 = \phi_3(t_i) = \text{Qwen-1.5B}_{\text{rhetoric}}(t_i)$, highlighting key linguistic markers of sarcasm.

**Visual Specialists.** For the visual component $v_i$, three visual specialist agents are employed:

- **Facial Expression Agent (ViT-FER):** Recognizes facial emotions with a Vision Transformer: $z_4 = \phi_4(v_i) = \text{ViT-FER}(v_i) \in \mathbb{R}^7$, outputting a 7D vector (e.g., happy, sad, angry).

- **Image Summarization Agent (BLIP-2):** Generates natural language scene descriptions: $z_5 = \phi_5(v_i) = \text{BLIP-2}(v_i) \rightarrow$ natural language description, providing high-level understanding.

- **Scene Text Agent (OCR-2.0):** Extracts embedded text in the image: $z_6 = \phi_6(v_i) = \text{OCR-2.0}(v_i) \rightarrow$ extracted text sequence, which may be crucial for sarcasm.

Each activated agent produces a structured output $z_k$ (information and confidence scores). Parallel execution ensures computational efficiency and precise analysis for complex sarcasm understanding.

## 2.5 Result Integration

After all activated agents have processed their respective sub-tasks, the Commander module $\mathcal{C}$ is responsible for integrating the intermediate outputs $\{z_k\}_{k \in \mathcal{A}_{\text{active}}}$ and making the final sarcasm prediction.

**Lightweight Encoder-Based Commander.** For encoder-based commanders (e.g., multi-modal BERT), we concatenate the outputs of all activated agents and project them through a learned fusion and classification head:

$$\mathbf{h}_{\text{fused}} = \text{concat}([z_k \text{ for } k \in \mathcal{A}_{\text{active}}]) \tag{6}$$

$$\mathbf{h}_{\text{context}} = \text{BERT}([\text{CLS}; \mathbf{h}_{\text{fused}}; \text{SEP}]) \tag{7}$$

$$\hat{y} = \text{softmax}(\mathbf{W}_{\text{out}}\mathbf{h}_{\text{context}} + \mathbf{b}_{\text{out}}) \tag{8}$$

where $\mathbf{W}_{\text{out}} \in \mathbb{R}^{2 \times d_{\text{hidden}}}$ and $\mathbf{b}_{\text{out}} \in \mathbb{R}^2$ are trainable parameters for binary classification.

**LLM Commander.** For moderately-sized language model commanders (e.g., DeepSeek-VL-7B) and large-scale LLM-based commanders (e.g., GPT-4o and Gemini Pro), all agent outputs are presented as a comprehensive structured prompt, and the LLM predicts both the sarcasm label and explanation:

$$\mathbf{p}_{\text{input}} = \text{Template}(\{z_k\}_{k \in \mathcal{A}_{\text{active}}}) \tag{9}$$

$$\hat{y} = \mathcal{C}_{\text{LM}}(\mathbf{p}_{\text{input}}) \rightarrow \{\text{Sarcastic}, \text{Non-sarcastic}\} \tag{10}$$

This three-stage architecture ensures that our Commander-GPT framework maintains both the specialized expertise of individual agents and the strategic coordination capabilities of military command structures, leading to more accurate and interpretable sarcasm detection results.

# 3 Experiments

In this section, we conduct comprehensive experiments on two widely-used multimodal sarcasm detection benchmarks, MMSD (Cai et al., 2019) and MMSD 2.0 (Qin et al., 2023).

## 3.1 Datasets

**MMSD:** The MMSD dataset is a benchmark for multimodal sarcasm detection, comprising paired textual and visual data collected from Twitter. Each example consists of a tweet and its associated image, with ground-truth sarcasm annotations. The textual content frequently features subtle or implicit sarcasm, while the accompanying images provide additional context, making the task especially challenging for both unimodal and multimodal models.

**MMSD 2.0:** MMSD 2.0 is an enhanced and extended version of MMSD, designed to support more robust evaluation of multimodal sarcasm detection systems. Compared to its predecessor, MMSD 2.0 significantly increases both the diversity of visual content and the quality of text-image alignment. It introduces more challenging cases where understanding sarcasm requires reasoning over complex interactions between modalities. Key statistics and properties of the two datasets are summarized in Table 1.

## 3.2 Experimental Settings

**Implementation Details.** All experiments were conducted on a server equipped with two NVIDIA RTX 4090 GPUs and 256GB RAM. Commander-GPT was implemented using PyTorch, HuggingFace Transformers, and the OpenMMLab toolkit. For supervised components (e.g., the BERT-based commander and the routing classifier), we fine-tuned for 10 epochs (approximately 12 hours in total). We used the AdamW

Table 1: Statistics of the MMSD and MMSD 2.0 datasets.

| Dataset | Train | Validation | Test | Sarcastic | Non-sarcastic | Source |
|---------|-------|-----------|------|-----------|---------------|--------|
| MMSD | 19,816 | 2,410 | 2,409 | 10,560 | 14,075 | Twitter |
| MMSD 2.0 | 19,816 | 2,410 | 2,409 | 11,651 | 12,980 | Twitter |

optimizer with an initial learning rate of $2 \times 10^{-5}$, batch size of 64, maximum sequence length of 512, and weight decay of 0.01. A linear warm-up and decay scheduler was applied. Early stopping was triggered if the validation F1 score did not improve for 3 consecutive epochs. Unless otherwise stated, all hyperparameters were tuned on the validation set. For LLM commanders (e.g., GPT-4o, Gemini Pro, DeepSeek-VL), we performed inference only without parameter fine-tuning.

**Baselines and Commander Configurations.** We compare Commander-GPT against five representative prompting and reasoning baselines:

- **Plan-and-Solve** (Wang et al., 2023): A pipeline prompting strategy that first plans the solution steps and then solves each subproblem sequentially.

- **Zero-shot CoT** (Kojima et al., 2022): A zero-shot chain-of-thought prompting method that enables step-by-step reasoning without requiring labeled demonstrations.

- **Generated Knowledge Prompting** (Liu et al., 2021): A method that augments the input with external knowledge generated by a language model to enhance reasoning.

- **Automatic Prompt Engineer** (Zhou et al., 2022): A technique that automatically searches for and optimizes prompt templates to maximize downstream task performance.

- **S$^3$ Agent** (Wang et al., 2024a): A multi-agent coordination framework for complex task decomposition and solution synthesis.

- **DMSD-CL** (Jia et al., 2024): A framework with contrastive learning that mitigates biased textual factors for robust OOD generalization, using counterfactual data augmentation to generate positive samples with dissimilar word biases and negative samples with similar word biases.

- **Cross-ToT** (Ranaldi et al., 2024): A cross-lingual Tree-of-Thoughts method that aligns reasoning across languages through a self-consistent cross-lingual prompting mechanism, enabling multi-step reasoning in different languages and achieving state-of-the-art performance with fewer interactions.

- **Graph of Thoughts** (Besta et al., 2024): A prompting framework that models LLM outputs as a graph with thoughts as nodes and dependencies as edges, enabling combination, distillation and feedback across thought networks, improving reasoning and supporting extensible thought transformations.

We further evaluate Commander-GPT under seven commander configurations, ranging from lightweight encoder-based commanders (BERT), to four medium-sized autoregressive LLMs (Yi-VL (6B) (Young et al., 2024), DeepSeek-VL-Chat (7B) (Lu et al., 2024), Qwen-VL-Chat (9B) (Bai et al., 2023), and MiniCPM-V-2 (2.8B) (Hu et al., 2024)), and two large instruction-following LLMs (Gemini Pro and GPT-4o).

### 3.3 Inference Efficiency

As shown in Fig. 3, we further evaluated the inference time and resource consumption of different commanders. On an RTX 4090 GPU with 256GB CPU memory, the BERT commander achieved an average latency of 0.12s per sample with a peak GPU memory usage of approximately 2.8 GB. For Yi-VL, DeepSeek-VL-Chat, Qwen-VL-Chat, and MiniCPM-V2, the average latencies were 0.45s, 0.68s, 0.52s, and 0.30s, with memory consumption of 13.8 GB, 32 GB, 10.5 GB, and 3.2 GB, respectively. In contrast, Gemini Pro and GPT-4o

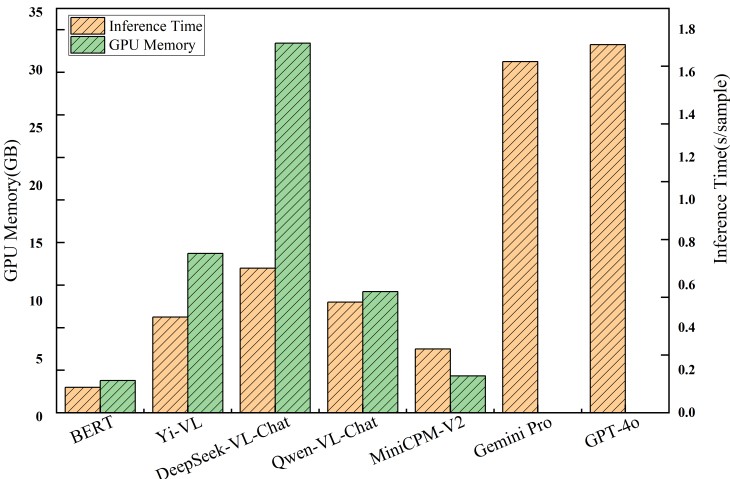

Figure 3: Inference efficiency

showed average latencies of 1.65s and 1.85s. Since Gemini Pro and GPT-4o are closed-source commercial models, our experiments were conducted via API calls, making it infeasible to report their GPU memory usage. Overall, while large LLM commanders deliver the best performance, their inference costs increase substantially; in contrast, small and medium models provide a more favorable trade-off between efficiency and accuracy.

## 3.4 Main results

Table 2 summarizes the performance of different commander architectures at three scales: lightweight encoder-based, small open-source LLMs, and SoTA proprietary LLMs. For all experiments, our routing-based method achieves the highest F1 scores and provides significant improvements over corresponding baselines.

**Lightweight Encoder-Based Commander.** The BERT+ViT-based commander represents the lightweight solution with minimal model size (110M parameters). Compared to direct fine-tuning, our routing method achieves substantial absolute gains: on MMSD, F1 improves from 80.8 to 86.7 (**+5.9**), and on MMSD 2.0 from 77.2 to 85.8 (**+8.6**). This corresponds to relative improvements of 7.3% and 11.1%, respectively. The results indicate that even for parameter-efficient models with limited multimodal reasoning ability, our approach enables more effective use of learned features, significantly boosting both overall accuracy and robustness.

**Small Autoregressive LLMs Commander.** For the small open-source LLMs group (Yi-VL, DeepSeek-VL-Chat, Qwen-VL-Chat, MiniCPM-V2), our method consistently outperforms all competitive prompting and multi-agent baselines. On MMSD 2.0, F1 improvements are especially pronounced: **+23.6** (Yi-VL), **+8.5** (DeepSeek-VL-Chat), **+10.3** (Qwen-VL-Chat), and **+22.8** (MiniCPM-V2). For instance, Qwen-VL-Chat reaches 68.9 F1 versus 58.6 for the best prior method, and MiniCPM-V2 rises from 49.2 to 72.0 F1. These results suggest that our approach is highly effective at extracting complementary information from multiple sub-agents in smaller models, compensating for their individual limitations.

**SoTA LLMs Commander.** For the strongest closed-source models (Gemini Pro, GPT-4o), the routing approach continues to yield improvements, though the margins are reduced compared to smaller models due to the high baseline performance. On MMSD 2.0, Gemini Pro improves from 67.4 to 71.9 F1 (**+4.5**), and GPT-4o increases from 73.2 to 76.5 F1 (**+3.3**). This demonstrates that even state-of-the-art models,

which already possess advanced multi-modal capabilities, benefit from structured subtask decomposition and information routing.

**Further Discussion.** These results highlight that Commander-GPT's effectiveness stems not only from numerical improvements but also from its novel design of structured task decomposition and routing. Unlike end-to-end models that implicitly handle sarcasm detection as a monolithic task, our framework explicitly separates the process into six cognitively meaningful subtasks. This design distributes complex reasoning to specialized expert agents, reducing the burden on any single model. The router then selectively activates only those agents most relevant for a given input, filtering out unnecessary noise. This "division of labor" enables agents to contribute complementary signals, which the commander integrates to detect subtle contradictions between text and vision. Consequently, the approach yields more robust and interpretable improvements over prior methods.

### 3.5 Comparison of Commander Models

Fig. 10 (see App. A4) presents the F1 scores of various commander models on the MMSD and MMSD 2.0 datasets. The results reveal significant performance differences among LLM backbones. BERT achieves the highest F1 scores (86.7 on MMSD, 85.8 on MMSD 2.0), primarily due to its fine-tuned nature on the sarcasm detection task. In contrast, mainstream vision-language models, such as Yi-VL (59.1/55.3) and DeepSeek-VL (61.1/60.5), perform substantially worse, likely due to their weaker language modeling capability or suboptimal adaptation to sarcasm-rich domains. Larger multimodal models like Qwen-VL, MiniCPM-V2, Gemini Pro, and GPT-4o show a clear performance hierarchy: GPT-4o delivers the strongest results among MLLMs (81.4/76.5), benefiting from both superior language modeling and robust multi-modal reasoning. Overall, these results highlight the importance of both model architecture and scale for effective sarcasm detection, and demonstrate that Commander-GPT can flexibly leverage a wide range of LLMs, with significant performance gain observed as model quality improves.

**Further Discussion** Notably, different commander models demonstrate distinct advantages within the Commander-GPT framework. The lightweight BERT commander, once fine-tuned, can effectively exploit structured subtask representations, allowing it to outperform some larger multimodal models. This underscores that model scale alone does not determine success—structured input and task specialization are equally crucial. For large models such as GPT-4o and Gemini Pro, while the relative gains are smaller, they remain significant compared to direct reasoning baselines. This indicates that even powerful general-purpose models benefit from explicit task decomposition and routing. Such findings reinforce the universality of Commander-GPT: it enhances task adaptability for lightweight models while simultaneously strengthening robustness and interpretability for large models.

### 3.6 Ablation Study

Table 3 and Table 4 present the ablation results on the MMSD and MMSD 2.0 datasets, respectively. Across all models, removing any single sub-task module leads to a noticeable drop in both F1 and accuracy (%), confirming the necessity of each component. For all three models, the largest relative performance degradation occurs when either *Rhetorical Device Recognition* or *Context Modeling* is ablated. For example, on MMSD, removing Rhetorical Device Recognition results in a 24.7% F1 decrease for DeepSeek-VL (from 61.1% to 27.9%), a 7.2% drop for MiniCPM-V2, and a 5.2% decrease for GPT-4o. Removing Context Modeling leads to a 6.9% F1 drop for DeepSeek-VL, 0.1% for MiniCPM-V2, and 2.5% for GPT-4o. On MMSD 2.0, similar trends are observed, with the ablation of Rhetorical Device Recognition causing up to 29.1% relative F1 drop for DeepSeek-VL, and 3.5% for GPT-4o.

For other sub-tasks such as Sentiment Analysis, Facial Expression Recognition, Image Summarization, and Scene Text Recognition, the impact is present but less pronounced, with typical F1 reductions ranging from 0.2% to 1.1% for GPT-4o, and 0.5% to 1.3% for MiniCPM-V2. All models achieve their best F1 and accuracy when all six modules are integrated, validating the effectiveness and complementarity of the full routing strategy. Overall, these results demonstrate that multi-dimensional sub-task collaboration is critical for robust multi-modal sarcasm detection, with context and rhetorical information providing the most substantial performance gains.

Table 2: Main Experimental Results. The **bolded** numbers indicate the best performance. Additionally, we calculated the increase in the F1 score.

| Model | Parameters | Method | MMSD | | | | MMSD 2.0 | | | |
|---|---|---|---|---|---|---|---|---|---|---|
| | | | F1 | Acc. | Pre. | Rec. | F1 | Acc. | Pre. | Rec. |
| **Lightweight Encoder-Based Commander** | | | | | | | | | | |
| BERT+ViT | 110M | BERT (Fine-tuned) | 80.8 | 82.6 | 81.2 | 80.1 | 77.2 | 79.7 | 77.4 | 75.8 |
| | | **Ours** | **86.7 (5.9↑)** | **90.1** | **87.4** | **86.1** | **85.8 (8.6↑)** | **87.4** | **86.1** | **85.3** |
| **Small Autoregressive LLMs Commander** | | | | | | | | | | |
| Yi-VL | 6B | Zero-shot CoT | 45.8 | 59.2 | 51.4 | 41.4 | 6.1 | 16.2 | 6.0 | 6.3 |
| | | Automatic Prompt Engineer | 49.4 | 53.3 | 45.0 | 54.7 | 4.3 | 14.2 | 4.2 | 4.5 |
| | | Plan-and-Solve | 51.3 | 62.6 | 56.2 | 47.2 | 4.4 | 9.2 | 4.0 | 4.8 |
| | | Generated Knowledge Prompting | 52.7 | 55.7 | 47.5 | 59.2 | 4.3 | 11.2 | 4.0 | 4.6 |
| | | $S^3$ Agent | 52.3 | 41.4 | 39.6 | **77.0** | 31.7 | 54.3 | 44.4 | 24.6 |
| | | DMSD-CL | 22.9 | 46.0 | **84.2** | 13.2 | - | - | - | - |
| | | Cross-ToT | 48.2 | 51.7 | 52.0 | 44.9 | 37.1 | 47.8 | 46.7 | 30.8 |
| | | GoT | 28.3 | 55.1 | 42.4 | 21.3 | 27.4 | 41.6 | 41.6 | 20.4 |
| | | **Ours** | **59.1 (6.4↑)** | **70.8** | 71.2 | 50.5 | **55.3 (18.2↑)** | **67.9** | **69.0** | **46.1** |
| DeepSeek-VL-Chat | 7B | Zero-shot CoT | 54.4 | 65.3 | 60.1 | 49.8 | 48.8 | 62.8 | 59.8 | 41.3 |
| | | Automatic Prompt Engineer | 55.1 | 68.0 | 66.4 | 47.6 | 46.6 | 62.3 | 59.7 | 38.2 |
| | | Plan-and-Solve | 54.9 | 61.9 | 54.3 | 55.5 | 48.3 | 63.0 | 60.5 | 40.2 |
| | | Generated Knowledge Prompting | 55.6 | 56.8 | 48.7 | **65.0** | 27.5 | 58.1 | 53.8 | 18.5 |
| | | $S^3$ Agent | 59.7 | 45.3 | 43.1 | 59.7 | 52.0 | **64.9** | **63.3** | 44.1 |
| | | DMSD-CL | 20.7 | 44.5 | **76.3** | 12.0 | - | - | - | - |
| | | Cross-ToT | 52.4 | 60.0 | 52.1 | 52.7 | 52.3 | 59.6 | 53.2 | 51.5 |
| | | GoT | 50.0 | 50.1 | 43.0 | 59.8 | 50.3 | 49.5 | 43.6 | 59.3 |
| | | **Ours** | **61.1 (1.4↑)** | **69.4** | 64.9 | 57.7 | **60.5 (6.2↑)** | 46.7 | 44.4 | **94.7** |
| Qwen-VL-Chat | 9B | Zero-shot CoT | 66.4 | 66.0 | 56.5 | 80.7 | 33.6 | 40.0 | 32.0 | 35.3 |
| | | Automatic Prompt Engineer | 64.5 | 60.5 | 51.6 | 86.0 | 33.3 | 40.4 | 32.1 | 34.5 |
| | | Plan-and-Solve | 59.4 | 64.8 | 57.2 | 61.8 | 34.8 | 38.7 | 32.1 | 38.0 |
| | | Generated Knowledge Prompting | 50.9 | 67.1 | **64.7** | 40.9 | 28.0 | 38.9 | 38.4 | 37.6 |
| | | $S^3$ Agent | 68.1 | 67.5 | 57.6 | 83.3 | 58.6 | 63.3 | 57.0 | 60.4 |
| | | DMSD-CL | 32.7 | 44.5 | 31.4 | 34.2 | - | - | - | - |
| | | Cross-ToT | 59.8 | 67.9 | 62.5 | 57.3 | 62.7 | 57.2 | 50.2 | 83.4 |
| | | GoT | 58.3 | 57.7 | 57.1 | 59.5 | 59.5 | 60.5 | 53.3 | 67.4 |
| | | **Ours** | **69.3 (1.2↑)** | **68.2** | 58.0 | 86.2 | **68.9 (6.2↑)** | **67.7** | **58.8** | 83.2 |
| MiniCPM-V2 | 2.8B | Zero-shot CoT | 61.4 | 68.1 | 62.1 | 60.7 | 33.0 | 51.6 | 40.9 | 27.7 |
| | | Automatic Prompt Engineer | 63.2 | 69.2 | 63.1 | 63.4 | 31.2 | 48.0 | 36.2 | 27.4 |
| | | Plan-and-Solve | 60.9 | 66.0 | 58.5 | 63.5 | 37.7 | 46.8 | 38.0 | 37.4 |
| | | Generated Knowledge Prompting | 63.2 | 67.5 | 59.9 | 66.8 | 47.5 | 47.3 | 41.6 | 55.3 |
| | | $S^3$ Agent | 60.8 | 66.6 | 59.6 | 62.0 | 49.2 | 62.9 | 60.0 | 41.8 |
| | | DMSD-CL | 46.4 | 47.5 | 43.1 | 50.3 | - | - | - | - |
| | | Cross-ToT | 35.3 | 56.4 | 46.4 | 28.6 | 37.9 | 56.1 | 48.5 | 31.1 |
| | | GoT | 57.9 | 42.1 | 41.6 | **95.4** | 59.2 | 43.3 | 42.9 | **95.5** |
| | | **Ours** | **72.5 (9.3↑)** | **74.5** | 65.9 | 80.8 | **72.0 (12.8↑)** | **73.9** | **66.8** | 78.1 |
| **SoTA LLMs Commander** | | | | | | | | | | |
| Gemini Pro | 600B | Zero-shot CoT | 67.2 | 71.4 | 62.8 | 70.5 | 59.4 | 61.1 | 53.9 | 66.1 |
| | | Automatic Prompt Engineer | 68.7 | 72.8 | 64.1 | 73.9 | 60.0 | 60.4 | 53.1 | 68.1 |
| | | Plan-and-Solve | 66.3 | 70.2 | 62.5 | 71.8 | 53.7 | 56.5 | 49.6 | 58.5 |
| | | Generated Knowledge Prompting | 69.5 | 73.6 | 66.3 | 74.2 | 65.1 | 61.7 | 53.6 | **82.8** |
| | | $S^3$ Agent | 70.1 | 74.9 | 67.8 | 75.7 | 67.4 | 66.5 | 58.0 | 80.3 |
| | | DMSD-CL | 52.6 | 53.7 | 48.2 | 57.9 | - | - | - | - |
| | | Cross-ToT | 67.5 | 66.6 | 66.0 | 69.0 | 59.5 | 59.3 | 60.6 | 58.3 |
| | | GoT | 70.5 | 71.8 | **75.4** | 66.2 | 66.4 | 66.0 | **70.8** | 62.6 |
| | | **Ours** | **73.8 (3.3↑)** | **76.2** | 70.1 | **78.3** | **71.9 (4.5↑)** | **73.8** | 68.4 | 76.7 |
| GPT-4o | 300B | Zero-shot CoT | 74.2 | 78.5 | 71.3 | 77.4 | 68.9 | 74.2 | 65.8 | 72.6 |
| | | Automatic Prompt Engineer | 75.8 | 79.1 | 72.9 | 78.9 | 70.3 | 75.6 | 67.1 | 73.8 |
| | | Plan-and-Solve | 73.6 | 77.8 | 70.2 | 77.1 | 67.4 | 73.1 | 64.2 | 71.0 |
| | | Generated Knowledge Prompting | 76.3 | 79.7 | 73.5 | 79.2 | 71.8 | 76.4 | 68.9 | 75.1 |
| | | $S^3$ Agent | 77.9 | 80.2 | 74.6 | 81.5 | 73.2 | 77.8 | 70.3 | 76.4 |
| | | DMSD-CL | 56.8 | 44.8 | **91.2** | 41.1 | - | - | - | - |
| | | Cross-ToT | 71.9 | 72.4 | 78.7 | 66.2 | 68.5 | 62.3 | **95.5** | 53.5 |
| | | GoT | 74.7 | 77.4 | 76.7 | 72.9 | 71.4 | 73.2 | 76.7 | 66.9 |
| | | **Ours** | **81.4 (3.5↑)** | **83.1** | 78.9 | **84.2** | **76.5 (3.3↑)** | **79.7** | 74.6 | **79.4** |

## 3.7 Subtask Invocation Frequency Analysis

Fig. 4 shows the invocation frequency for each subtask agent. **Context Modeling**, **Sentiment Analysis**, and **Image Summarization** are triggered in all cases (2,409 times each), reflecting their necessity for every input. By contrast, **Rhetorical Device Recognition**, **Facial Expression Recognition**, and **Scene Text**

Table 3: Ablation study on the MMSD dataset.

| Ablation Setting | MiniCPM-V2 | | DeepSeek-VL | | GPT-4o | |
|---|---|---|---|---|---|---|
| | F1 | Acc. | F1 | Acc. | F1 | Acc. |
| w/o Context Modeling | 72.4 | 73.9 | 56.9 | 70.9 | 78.9 | 81.2 |
| w/o Sentiment Analysis | 71.3 | 72.2 | 60.2 | 71.9 | 79.1 | 81.4 |
| w/o Rhetorical Device Recognition | 66.0 | 72.4 | 27.9 | 63.9 | 77.2 | 79.8 |
| w/o Facial Expression Recognition | 71.2 | 72.2 | 41.7 | 66.8 | 78.5 | 81.0 |
| w/o Image Summarization | 71.4 | 73.1 | 43.8 | 67.3 | 78.8 | 81.3 |
| w/o Scene Text Recognition | 69.4 | 71.5 | 28.8 | 63.4 | 77.8 | 80.1 |
| **Full Model** | **72.5** | **74.5** | **61.1** | **69.4** | **81.4** | **83.1** |

Table 4: Ablation study on the MMSD 2.0 dataset.

| Ablation Setting | MiniCPM-V2 | | DeepSeek-VL | | GPT-4o | |
|---|---|---|---|---|---|---|
| | F1 | Acc. | F1 | Acc. | F1 | Acc. |
| w/o Context Modeling | 72.1 | 73.6 | 59.4 | 43.1 | 74.1 | 77.2 |
| w/o Sentiment Analysis | 71.2 | 71.6 | 59.0 | 42.9 | 74.3 | 77.4 |
| w/o Rhetorical Device Recognition | 67.9 | 72.3 | 60.0 | 42.5 | 73.8 | 77.0 |
| w/o Facial Expression Recognition | 71.5 | 72.4 | 59.7 | 43.2 | 74.0 | 77.1 |
| w/o Image Summarization | 71.9 | 73.3 | 59.5 | 43.0 | 74.2 | 77.3 |
| w/o Scene Text Recognition | 67.9 | 62.2 | 60.2 | 43.1 | 73.2 | 76.4 |
| **Full Model** | **72.0** | **73.9** | **60.5** | **46.7** | **76.5** | **79.7** |

**Recognition** are conditionally invoked, with call counts of 1,479, 1,285, and 935, respectively. The lower frequencies for the latter agents are due to the absence of rhetorical devices, facial expressions, or scene text in a significant proportion of samples.

This distribution highlights two important findings. First, the frequency of subtask invocation is closely linked to the nature of the input data (i.e., whether textual or visual clues are present). Second, agents such as Facial Expression Recognition and Scene Text Recognition—though not always activated—capture critical cues for sarcasm detection when relevant. These results further support the rationality of designing dedicated subtasks and specialized agents, as each contributes unique information to the overall multi-agent sarcasm understanding framework.

### 3.8 Out-of-Distribution Results

To evaluate the robustness and generalization of Commander-GPT, we conducted out-of-distribution sarcasm detection experiments on the SemEval 2018 Task 3 dataset. We selected two LLMs as the commander: the open-source MiniCPM-V2, and the SoTA model Claude-3.5 (serving as a new commander).

Fig. 5 reports the F1 and Acc. scores of all methods. On MiniCPM-V2, Commander-GPT achieves an F1 of 73.6% and an Acc. of 72.6%, outperforming all prompt-based and agent-based baselines. Specifically, compared to the strongest baseline ($S^3$ Agent, F1 69.5%, Acc. 67.5%), Commander-GPT brings improvements of 4.1% and 5.1%, respectively.

When using Claude 3.5 as the commander, Commander-GPT still maintains the leading performance, achieving 62.1% F1 and 51.7% Acc. This exceeds the best baseline (Zero-shot CoT, F1 59.8%, Acc. 46.8%) by 2.3% and 4.9%, respectively. It is worth noting that the performance boost remains consistent even when switching to a more powerful closed-source model as the commander, demonstrating the flexibility and model-agnostic nature of our framework.

Overall, these results confirm that Commander-GPT can robustly generalize to new domains and leverage the advantages of different backbone LLMs. Whether with open-source or SOTA commercial models as the commander, our framework consistently delivers state-of-the-art results in sarcasm detection.

### 3.9 The Impact of The Number of Subtasks

From the experimental results in Fig. 6, it can be seen that as the number of submodules increases, the model's F1 score on the MMSD and MMSD 2.0 datasets follows a three-stage trend of "slow start, rapid rise, then gentle plateau with a final slight boost." The first two basic modules—Context Modeling and

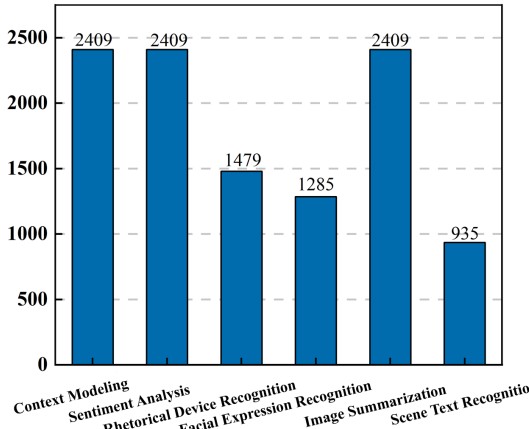

Figure 4: Agent call frequency on the MMSD 2.0 dataset.

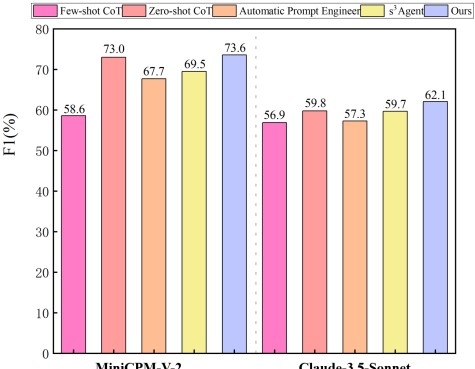 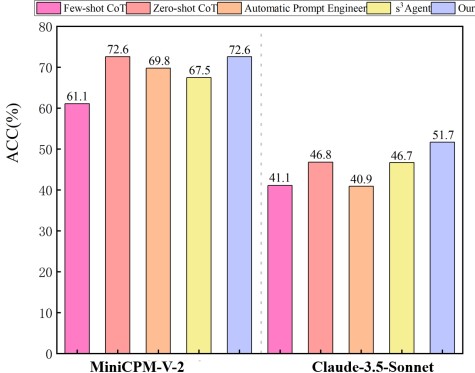

Figure 5: The experimental results of MiniCPM-V-2 and Claude-3 on SemEval 2018 Task 3 were statistically evaluated.

Sentiment Analysis—bring only limited gains until the introduction of Image Summarization and Scene Text Recognition triggers a substantial jump, with Scene Text Recognition especially filling the gap between textual and visual information. Adding Rhetorical Device Recognition and Facial Expression Recognition thereafter still yields improvements, but with diminishing marginal returns. On MMSD 2.0, thanks to higher data quality, the modules cooperate more smoothly and deliver steadier performance gains. Therefore, under resource constraints, priority should be given to Image Summarization and Scene Text Recognition, followed by Rhetorical Device Recognition and Facial Expression Recognition, while cross-modal fusion strategies can be explored to further unlock the potential of later modules.

### 3.10 Routing Score Visualization

To further interpret the decision mechanism of Commander-GPT, we visualize the routing scores assigned to each subtask agent for representative samples from the MMSD 2.0 dataset. Fig. 7 presents a heatmap where each column corresponds to an input sample and each row corresponds to one of the six subtask agents. The color intensity indicates the routing score (i.e., the normalized weight assigned by the router to each agent).

The heatmap reveals clear patterns in the agent assignment. For samples containing explicit rhetorical devices, the router assigns high scores to the *Rhetorical Device Recognition* agent. Samples with prominent facial expressions or scene text result in higher weights for the *Facial Expression Recognition* and *Scene Text Recognition* agents, respectively. In contrast, samples lacking such cues rely more on the *Context*

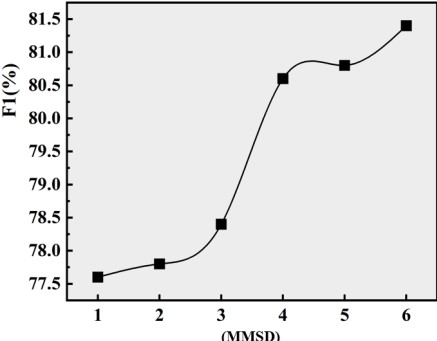 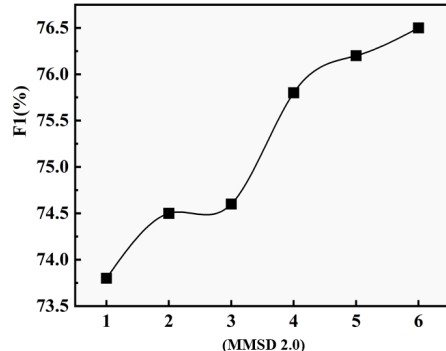

Figure 6: F1 score variation curves for the MMSD and MMSD 2.0 datasets with varying numbers of sub-modules, where the modules are added in the following order: Context Modeling, Sentiment Analysis, Image Summarization, Scene Text Recognition, Rhetorical Device Recognition, and Facial Expression Recognition.

*Modeling* and *Sentiment Analysis* agents. This dynamic and adaptive score distribution demonstrates that the router effectively identifies and leverages the most relevant subtasks for each input, thereby improving interpretability and robustness.

Overall, the routing score visualization provides direct evidence that the proposed multi-agent architecture can selectively integrate different information sources based on the input characteristics, supporting both the transparency and effectiveness of our approach.

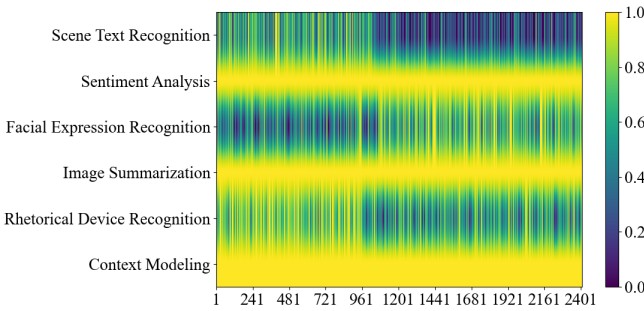

Figure 7: Heatmap of subtask agent counts on the MMSD 2.0 dataset.

### 3.11 Error Analysis

In the first three examples, Commander-GPT demonstrates its strength in multimodal sarcasm detection by accurately identifying sarcastic or non-sarcastic content. The first example, "when my best friend tells me he got full marks in the exam," contains sarcasm, which GPT-4o and Claude-4 both fail to detect, labeling it as non-sarcastic. However, Commander-GPT successfully recognizes the sarcastic intent by utilizing a combination of subtasks, including Context Modeling, Sentiment Analysis, Facial Expression Recognition, Scene Text Recognition, and Image Summarization. This highlights Commander-GPT's ability to capture nuanced contextual cues across both text and images, which are crucial for accurate sarcasm detection. Similarly, in the second example, "this is reality...lol," GPT-4o and Claude-4 misclassify it as non-sarcastic, but Commander-GPT correctly identifies it as sarcastic, again by leveraging multimodal cues and the combination of its subtasks. These successes illustrate the advantage of breaking down the task into subtasks that handle specific aspects of sarcasm, like emotional tone, context, and visual cues, enhancing the model's overall performance.

In the fourth example, "when you say I love you to a person you hate," Commander-GPT misclassifies the sarcastic input as non-sarcastic, while both GPT-4o and Claude-4 correctly classify it as sarcastic. This failure suggests a limitation in Commander-GPT's model, especially in scenarios where certain subtasks might fail to capture the sarcasm cues. For instance, despite the clear sarcastic nature of the statement, Commander-GPT might have missed critical contextual or emotional cues from the text or visual content. While the model effectively uses Context Modeling and Sentiment Analysis, it appears to misinterpret the sarcasm in this specific scenario, likely due to a failure in one of the subtasks. This highlights the potential weakness of multimodal models when handling certain kinds of sarcasm that do not have clear visual or textual cues, emphasizing the need for further refinement of the task decomposition and routing strategy to better handle such cases.

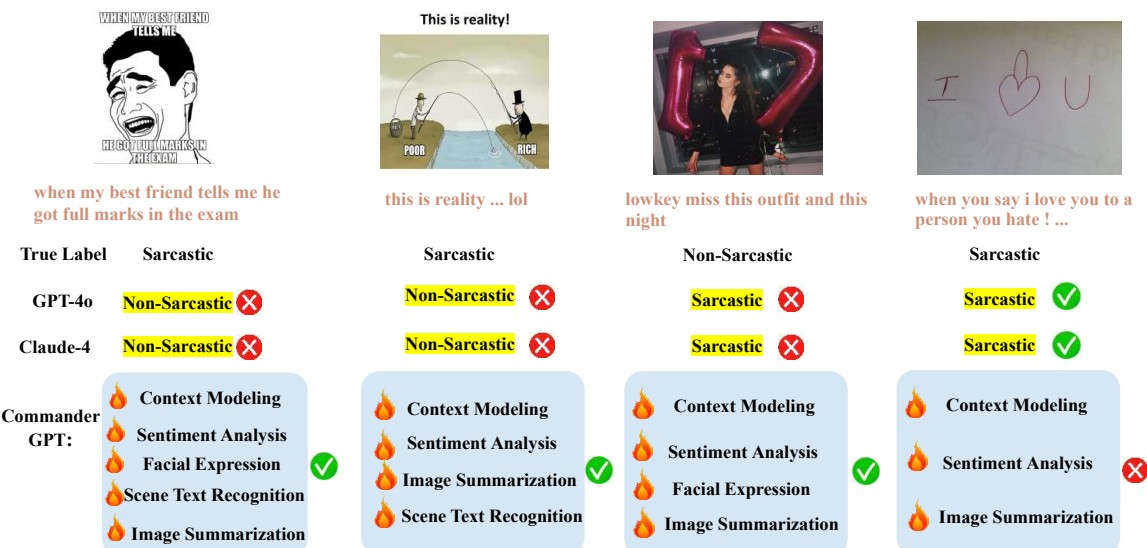

Figure 8: The figure shows four examples in a multimodal sarcasm detection task, along with the classification results of different models (GPT-4o, Claude-4, Commander-GPT) for these examples.

## 4 Conclusion

In this paper, we proposed Commander-GPT, a novel multi-agent routing framework for multimodal sarcasm detection. Our approach decomposes the complex sarcasm detection task into cognitively meaningful subtasks, dynamically routing each input to the most suitable specialist agent. The central commander integrates information from all subtask agents, enabling adaptive and fine-grained reasoning across both textual and visual modalities. Extensive experiments on multiple benchmarks demonstrate that Commander-GPT consistently achieves state-of-the-art performance. Further analyses confirm the effectiveness, robustness, and strong generalization ability of our framework. We believe this work offers new insights into explainable and modular design for sarcasm and sentiment analysis.

**Discussion on Hallucination.** A potential risk when applying large language models to sarcasm detection is the occurrence of hallucinations, where the model invents context, emotions, or visual details that are not supported by the input. For example, we observed cases where a commander LLM attributed sarcasm to an "angry facial expression" even though no face was present in the image, or generated an explanation that contradicted the literal text.

Our own case study (Fig. 9) further illustrates this issue. The input was an image of hamburgers and fries with the caption "feeding my abs nothing but the best quality beef." Humans can easily recognize the sarcastic intent here: the mention of "abs" contrasts ironically with the unhealthy fast food in the picture. However, the LLM produced hallucinations across multiple analysis steps:

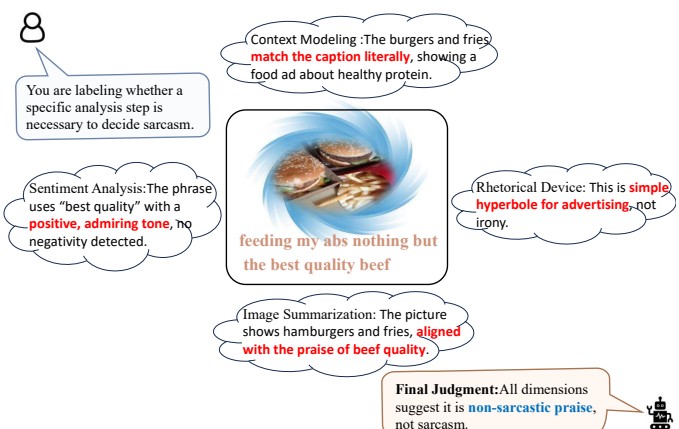

Figure 9: Commander-GPT Hallucination Case

- Context Modeling: it claimed the burgers and fries "match the caption literally," inventing an advertising context about healthy protein that does not exist.

- Sentiment Analysis: it highlighted "best quality" as a "positive, admiring tone," hallucinating a purely positive sentiment while ignoring the ironic undertone.

- Rhetorical Device: it labeled the statement as "simple hyperbole for advertising" rather than irony, a clear misinterpretation of the rhetorical device.

- Image Summarization: it concluded that the picture was "aligned with the praise of beef quality," overlooking the tension between junk food and the notion of feeding one's abs.

As shown in the figure, these hallucinations accumulated in the final stage, leading the model to incorrectly judge the overall message as "non-sarcastic praise." This example demonstrates how subtle pragmatic cues can be lost when the model hallucinates supportive evidence, producing confident but misleading explanations. Such hallucinations are particularly problematic in sarcasm detection, as the task already relies on subtle pragmatic cues and fine-grained multimodal alignment. While our modular design partially mitigates this issue by constraining each agent to a well-defined subtask, hallucinations can still emerge in the integration stage when the commander synthesizes final decisions. Future research could incorporate uncertainty estimation, factual consistency checks, or human-in-the-loop validation to reduce the impact of hallucinations in real-world deployment.

**Limitations.** Although Commander-GPT shows strong effectiveness in multimodal sarcasm detection, several limitations remain. First, *scalability* poses a challenge: increasing the number of agents or adopting larger commanders inevitably raises inference latency and memory consumption, which may hinder real-time or large-scale applications. While our analysis in §3.3 reports efficiency trade-offs, more systematic profiling is needed. Second, the framework *relies on external specialist models* (e.g., BLIP-2, OCR-2.0, RoBERTa, GPT-4o). This dependency makes stability and reproducibility sensitive to changes in upstream APIs or pretrained checkpoints. Finally, for *real-world deployment*, the current design is resource-intensive. Promising directions to make the framework lighter include agent pruning and parameter sharing across subtasks, compression and distillation of commanders, or integrating all subtasks into a unified but modular multimodal backbone. We view these as important next steps toward reducing resource consumption while maintaining interpretability.

**Acknowledgements.** This paper is supported by The National Social Science Fund of China (No. 25AYY001), Hong Kong RGC Theme-based Research Scheme (T45-401/22-N) and a grant under the scheme of Collaborative Research with World-leading Research Groups in The Hong Kong Polytechnic University (project no. G-SACF), Natural Science Foundation of Henan Province of China (242300421412), National Natural Science Foundation of China (No.62201572).

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

# A  Appendix

## A.1  Related Work

### A.1.1  Multimodal Large Language Models

Multimodal large language models (MLLMs) aim to unify language and vision understanding through joint pretraining and alignment across modalities. Early frameworks such as CLIP (Radford et al., 2021) and BLIP (Li et al., 2022) align vision and text representations through contrastive learning or caption supervision. These models laid the groundwork for subsequent generative MLLMs like Flamingo (Alayrac et al., 2022), MiniGPT-4 (Zhu et al., 2023), and LLaVA (Liu et al., 2023a), which integrate vision encoders with pretrained LLMs via lightweight adapters or projection layers. These models support open-ended visual question answering and caption generation. Qwen-VL (Bai et al., 2023), InternVL (Chen et al., 2024), and Emu2 (Sun et al., 2024) further explore multi-granular alignment, spatial grounding, and tool-augmented multimodal reasoning.

While MLLMs such as GPT-4o and Gemini have achieved strong performance in visual grounding and factual question answering, they consistently underperform on high-level pragmatic tasks like sarcasm and irony (Yao

et al., 2025; Yang et al., 2024). Sarcastic expressions often rely on rhetorical contradiction and contextual incongruity, which current models struggle to capture. Notably, Yao et al. (2025) proposed advanced prompting strategies, i.e., *graph-of-cue* and *bagging-of-cue* to improve sarcasm detection, but found them less effective than standard input–output approach. This suggests that increasing prompt complexity alone cannot overcome the reasoning limitations of monolithic models. In contrast, our proposed Commander-GPT framework explicitly decomposes sarcasm detection into specialized sub-tasks, each handled by expert agents under centralized coordination, enabling more effective and interpretable multimodal sarcasm understanding.

### A.1.2   Multimodal Sarcasm Detection

Multimodal sarcasm detection is a complex task that requires the integration of linguistic, visual, and contextual information to resolve non-literal intent and pragmatic ambiguity. In recent years, researchers have explored a variety of neural architectures and multimodal fusion strategies to address the inherent challenges of this task.

Recent work has shifted toward more advanced architectures that better capture the nuanced interplay between modalities. For example, Wang et al. (2024a) proposed the $S^3$ Agent framework, which employs visual large language models and integrates multi-perspective analysis to enhance zero-shot multimodal sarcasm detection. Similarly, Tang et al. (2024) utilized generative MLLMs equipped with instruction templates and demonstration retrieval to improve the understanding of complex sarcastic cues. Aggarwal et al. (Aggarwal et al., 2024) designed a framework capable of processing multimodal input triples—including text, images, and descriptive captions—highlighting the benefit of incorporating multiple contextual sources.

To further enhance semantic understanding, researchers have explored the integration of external knowledge bases. KnowleNet (Yue et al., 2023), for instance, leverages ConceptNet to inject prior knowledge and assesses cross-modal semantic similarity at both sample and word levels. Other recent models, such as RCLMuFN (Wang et al., 2024b), introduce relational context learning and multi-path fusion networks to improve generalization and robustness in sarcasm detection.

Despite these advances, most existing methods either rely on monolithic model architectures or simple cross-modal fusion, which often fail to capture the composite and context-dependent nature of sarcasm. In this context, our proposed Commander-GPT addresses these challenges by decomposing multimodal sarcasm detection into cognitively meaningful sub-tasks, each handled by a specialized agent under centralized coordination. This design aims to improve the performance of multimodal sarcasm detection.

### A.2   Algorithm

The algorithm is shown in Alg. 1.

### A.3   Prompt Construction

For instruction-tuned LLMs (e.g., GPT-4o, DeepSeek VL), agent routing is formulated as a natural language inference task. For each $(x, T_k)$, we construct the prompt:

---

**Algorithm 1** Commander-GPT: Modular Multimodal Sarcasm Understanding

---

1: **Input:** Multimodal input $x = (t, v)$
2: **Output:** Sarcasm prediction $\hat{y}$

3: **1. Agent Routing:**
4: Obtain agent activation vector $[r_1(x), \ldots, r_K(x)]$ using routing scorer.
5: $\mathcal{A}_x \leftarrow \{A_k \mid r_k(x) = 1\}$

6: **2. Subtask Execution:**
7: **for all** activated agent $A_k \in \mathcal{A}_x$ **in parallel do**
8:     Compute subtask output $z_k = \phi_k(x)$
9: **end for**
10: Collect all outputs $\mathbf{z} = [z_k \mid A_k \in \mathcal{A}_x]$

11: **3. Result Integration and Final Decision:**
12: **if** commander is encoder-based **then**
13:     Fuse outputs: $\mathbf{h}_{\text{fused}} \leftarrow \text{concat}(\mathbf{z})$
14:     Compute hidden: $\mathbf{h}_{\text{context}} \leftarrow \text{BERT}([\text{CLS}; \mathbf{h}_{\text{fused}}; \text{SEP}])$
15:     Predict: $\hat{y} = \text{softmax}(\mathbf{W}_{\text{out}}\mathbf{h}_{\text{context}} + \mathbf{b}_{\text{out}})$
16: **else**
17:     Format prompt $\mathbf{p}_{\text{input}}$ from all $z_k$
18:     Predict: $\hat{y} = \mathcal{C}_{\text{LLM}}(\mathbf{p}_{\text{input}})$
19: **end if**

---

> **System:** You are a military commander analyzing intelligence to deploy specialized units for sarcasm detection.
> **Input:** Text: "{<t_i>}", Image: {<v_i>}
> **Task:** Determine which of the following units should be deployed:
>
> - "context_modeling": Analyze broader conversational context
>
> - "sentiment_analysis": Extract emotional polarity
>
> - "rhetorical_device": Identify irony, hyperbole, etc.
>
> - "facial_expression": Analyze facial emotions in image
>
> - "image_summarization": Describe visual content
>
> - "scene_text": Extract text from image
>
> **Output:** {"context_modeling": 0/1, "sentiment_analysis": 0/1, ...}

### A.4 Comparison of Commander Models

Fig. 10 (App. A4) presents the F1 scores of various commander models on the MMSD and MMSD 2.0 datasets. The results reveal significant performance differences among LLM backbones. BERT achieves the highest F1 scores (86.7 on MMSD, 85.8 on MMSD 2.0), primarily due to its fine-tuned nature on the sarcasm detection task. In contrast, mainstream vision-language models, such as Yi-VL (59.1/55.3) and DeepSeek-VL (61.1/60.5), perform substantially worse, likely due to their weaker language modeling capability or suboptimal adaptation to sarcasm-rich domains. Larger multimodal models like Qwen-VL, MiniCPM-V2, Gemini Pro, and GPT-4o show a clear performance hierarchy: GPT-4o delivers the strongest results among MLLMs (81.4/76.5), benefiting from both superior language modeling and robust multi-modal reasoning. Overall, these results highlight the importance of both model architecture and scale for effective

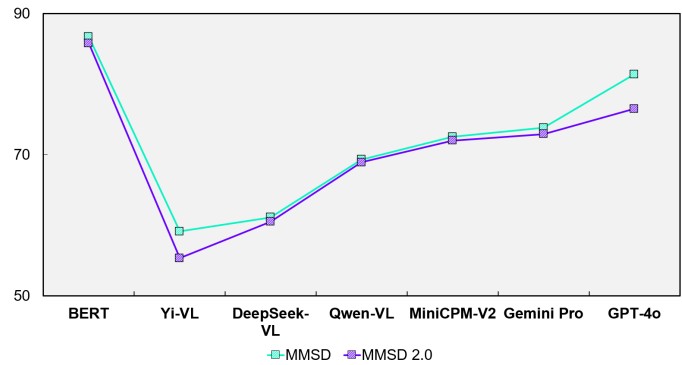

Figure 10: Comparison of commander models on the MMSD and MMSD 2.0 dataset.

sarcasm detection, and demonstrate that Commander-GPT can flexibly leverage a wide range of LLMs, with significant performance gain observed as model quality improves.

## A.5  Prompt Design

We reformulate agent routing as a binary decision task: given the text, image, and a candidate analysis step, the model outputs whether this step is necessary for sarcasm detection, in JSON format.

**System:** You are labeling whether a specific analysis step is necessary to decide sarcasm.
**User:**
TEXT: "{<t>}"
IMAGE: {<v>} (attached)

**TASK:** "{<T_k>}"  # e.g., "facial_expression", "scene_text", "image_summarization", "context_modeling", "sentiment_analysis", "rhetorical_device"

**Instruction:** Reply in JSON ONLY:
{"need": "Yes" or "No"}

