# OpenReview forum: "Commander-GPT: Dividing and Routing for Multimodal Sarcasm Detection"
_TMLR — Accepted by TMLR_

### Review · Reviewer_sW8L · 2025-08-27

**Summary Of Contributions:**

The paper introduces Commander-GPT, a modular multi-agent framework designed for multimodal sarcasm detection, inspired by military command theory. Instead of relying on a single large language model, the system decomposes sarcasm detection into six subtasks, context modeling, sentiment analysis, rhetorical device recognition, facial expression recognition, image summarization, and scene text recognition, each handled by specialized agents. A central commander model dynamically routes tasks to the most suitable agents, integrates their outputs, and makes the final sarcasm judgment. Experiments on MMSD and MMSD 2.0 benchmarks show that Commander-GPT outperforms state-of-the-art baselines by up to 11.7% in F1 score, demonstrating improved accuracy, robustness, and generalization.

**Audience:**

Yes

**Audience Explanation:**

TMLR’s audience would be interested because sarcasm detection is a challenging and unsolved problem in NLP, especially in multimodal settings. The paper introduces a novel modular framework, Commander-GPT, which not only advances performance but also provides insights into explainable, task-decomposed reasoning with LLMs. These contributions are valuable to researchers in language understanding, multimodal learning, and sentiment analysis, making the findings relevant to the community.

**Broader Impact Concerns:**

A potential concern is that applying sarcasm detection in sensitive areas, such as social media monitoring or surveillance, could result in misclassifications that cause misunderstandings or lead to unfair treatment of individuals.

**Claims And Evidence:**

Yes

**Claims Explanation:**

the claims are well supported by clear evidence. The paper justifies the need for subtask decomposition, details the Commander-GPT design, and evaluates it thoroughly on MMSD and MMSD 2.0 against strong baselines. Results consistently show significant F1 gains, and ablation plus out-of-distribution tests further confirm robustness. Together, these analyses convincingly support the authors’ claims.

**Requested Changes:**

1. I think the authors should also mention and discuss other multi-modality Sarcasm Detection approaches. From paper [1] Table 2 we could find more existing researches for this task which also delivered comparable performances. These work should be compared and discussed in this paper.

[1] Jia, Mengzhao, Can Xie, and Liqiang Jing. "Debiasing multimodal sarcasm detection with contrastive learning." Proceedings of the AAAI conference on artificial intelligence. Vol. 38. No. 16. 2024.

2. In Sec 3.3 and Sec 3.4, the authors are encouraged to add more discussion regarding why the proposed method could work better than the others towards the novel design. This discussion will help to enrich the insights of the experimental analysis section. For the current version it is only the description of the numbers without in-depth analysis.


3. Lack of the qualitative results. The authors are encouraged to enrich the qualitative results together with failure cases for different approaches on different LLMs. The failure cases can help the authors to understand the limitation of the proposed approach well.

4. Most of the baselines selected (i.e., Plan-and-Solve (Wang et al., 2023), Zero-shot CoT (Kojima et al., 2022), Generated Knowledge Prompting (Liu et al., 2021), Automatic Prompt Engineer (Zhou et al., 2022), and  S3Agent (Wang et al., 2024a)) are the works before 2024, which are not very recent. The authors are encouraged to make comparison with some recent techniques, e.g., [2] and [3].

[2] Ranaldi, Leonardo, et al. "A tree-of-thoughts to broaden multi-step reasoning across languages." Findings of the Association for Computational Linguistics: NAACL 2024. 2024.

[3] Besta, Maciej, et al. "Graph of thoughts: Solving elaborate problems with large language models." Proceedings of the AAAI conference on artificial intelligence. Vol. 38. No. 16. 2024.

5. I am wondering the hallucination of the LLMs on the sarcasm detection. Could the authors elaborate more on discussion regarding this direction?

---

> ### Author Response · Authors · 2025-09-30
>
> Dear Editor, Dear Reviewer,
>
> We have carefully revised the paper in response, with changes highlighted in red, and prepared a detailed point-by-point reply.
>
>
> **Q1. I think the authors should also mention and discuss other multi-modality Sarcasm Detection approaches. From paper [1] Table 2 we could find more existing researches for this task which also delivered comparable performances. These work should be compared and discussed in this paper.[1] Jia, Mengzhao, Can Xie, and Liqiang Jing. "Debiasing multimodal sarcasm detection with contrastive learning." Proceedings of the AAAI conference on artificial intelligence. Vol. 38. No. 16. 2024.**
>
> **Answer 1:** Thank you for pointing this out. In the revised version, we have extended the related work and experimental discussion sections to include more multimodal sarcasm detection methods, including the work of Jia et al. (2024) [1]. We now explicitly compare Commander-GPT with these approaches and discuss their relative advantages and limitations.
>
>
> **Q2. In Sec 3.3 and Sec 3.4, the authors are encouraged to add more discussion regarding why the proposed method could work better than the others towards the novel design. This discussion will help to enrich the insights of the experimental analysis section. For the current version it is only the description of the numbers without in-depth analysis.**
>
> **Answer 2:** We agree with the reviewer that the original discussion focused more on reporting numerical results. In the revised manuscript, we have enriched Sections 3.3 and 3.4 with additional analysis, emphasizing why our proposed design of modular subtask decomposition and dynamic routing contributes to superior performance compared to monolithic baselines. This provides a clearer explanation of the effectiveness of our framework beyond the numerical improvements.
>
>
> **Q3. Lack of the qualitative results. The authors are encouraged to enrich the qualitative results together with failure cases for different approaches on different LLMs. The failure cases can help the authors to understand the limitation of the proposed approach well.**
>
> **Answer 3:** Following the reviewer’s suggestion, we have incorporated more qualitative examples, including successful cases as well as failure cases across different commander models (see Sec. 3.11, Fig. 8). These examples not only demonstrate the strengths of Commander-GPT but also highlight its limitations, providing valuable insights for future research.
>
>
> **Q4. Most of the baselines selected (i.e., Plan-and-Solve (Wang et al., 2023), Zero-shot CoT (Kojima et al., 2022), Generated Knowledge Prompting (Liu et al., 2021), Automatic Prompt Engineer (Zhou et al., 2022), and S3Agent (Wang et al., 2024a)) are the works before 2024, which are not very recent. The authors are encouraged to make comparison with some recent techniques, e.g., [2] and [3].[2] Ranaldi, Leonardo, et al. "A tree-of-thoughts to broaden multi-step reasoning across languages." Findings of the Association for Computational Linguistics: NAACL 2024. 2024.[3] Besta, Maciej, et al. "Graph of thoughts: Solving elaborate problems with large language models." Proceedings of the AAAI conference on artificial intelligence. Vol. 38. No. 16. 2024.**
>
> **Answer 4:** We acknowledge that the baselines in the original submission were primarily pre-2024. In the revised version, we have added comparisons with recent state-of-the-art prompting frameworks, including Cross-ToT (Ranaldi et al., 2024) [2] and Graph of Thoughts (Besta et al., 2024) [3], as shown in Table 2. These additional comparisons further validate the competitiveness of Commander-GPT against the latest techniques.
>
> **Q5.  I am wondering the hallucination of the LLMs on the sarcasm detection. Could the authors elaborate more on discussion regarding this direction?**
>
>
> **Answer 5:**  We appreciate this valuable comment. In the revised version, we have added a dedicated discussion on hallucination risks in Section 4 (Discussion on Hallucination, see Fig. 9). We provide concrete case studies where LLMs generate misleading intermediate reasoning steps, analyze the causes, and suggest potential directions (e.g., uncertainty estimation, factual consistency checks) to mitigate such risks.

---

### Review · Reviewer_FhKc · 2025-09-06

**Summary Of Contributions:**

The paper introduces Commander-GPT, a modular multi-agent framework for multimodal sarcasm detection. It decomposes the sarcasm detection task into $6$ subtasks and the commander dynamically choose which of these subtasks should be carried out given the input. Empirical experiments on MMSD and MMSD 2.0 datasets are carried out to exhibit its improvement compared to the baselines.

**Audience:**

Yes

**Audience Explanation:**

- This paper proposes a pipeline, Commander-GPT, that employ different large language models to conduct $6$ subtasks that help sarcasm detection.
- The experiments demonstrate the efficacy of the proposed framework, and that each subtask contribute meaningful impact to the final detection.

**Broader Impact Concerns:**

None.

**Claims And Evidence:**

No

**Claims Explanation:**

- Optimization clarity: Eq. (2) states joint minimization over $\alpha,\theta_C$​, but the method trains a router via Eq. (3)–(4) with GPT-4o-distilled labels plus threshold gating at inference. Clarify whether $\alpha$ is jointly optimized with $\theta_C$​ or learned in a staged manner only.
- Agent dependence: Performance hinges on external specialists (BLIP-2, OCR-2.0, RoBERTa, etc.); robustness to swapping/weaker agents is not evaluated.
- Overhead: Multi-agent orchestration likely increases latency/memory; no runtime/resource comparison vs. single strong LLM baselines (e.g., fine-tuned Llama-3-8B). (Table 2 includes GPT-4o/Gemini Pro but not a monolithic fine-tuned open LLM.)

**Requested Changes:**

- In equation (1), I suppose the right-hand side should be indexed up to $K$?
- The curly brackets are missing in some expressions, e.g., second last line on page 3, second line on page 4, equation (1), etc. When using the curly brackets in tex, it should be companied by back slash.
- Please explicitly describe the training pipeline (router via (3)–(4), commander via (5)–(7)), and state whether any joint or end-to-end optimization of $\alpha$ and $\theta_C$​ is performed. If not, rephrase Eq. (2) or present a bilevel/alternating optimization view.

---

> ### Author Response · Authors · 2025-09-30
>
> Dear Editor, Dear Reviewer,
>
> On behalf of all authors, we sincerely thank you for your valuable comments on our manuscript “Commander-GPT: Dividing and Routing for Multimodal Sarcasm Detection” (Submission Number: 5338). We have carefully revised the paper in response, with changes highlighted in red, and prepared a detailed point-by-point reply. We believe these revisions have greatly improved the quality of the work, and we hope the revised version will meet the requirements for publication.
>
> Sincerely,
>
>
> **Q1. In equation (1), I suppose the right-hand side should be indexed up to K?**
>
> **Answer 1:** For Equation (1), we note your suggestion that the index on the right-hand side should extend to K. In the revised version, we have corrected this issue: the index in Equation (1) now properly extends to K, ensuring consistency of the expression.
>
>
> **Q2. The curly brackets are missing in some expressions, e.g., second last line on page 3, second line on page 4, equation (1), etc. When using the curly brackets in tex, it should be companied by back slash.**
>
> **Answer 2:** We have carefully checked the entire manuscript, in particular the second-to-last line on page 3, the second line on page 4, and the missing curly braces in Equation (1). We have added the correct formatting in these places. We ensured that all expressions involving curly braces now follow the correct syntax, and that each curly brace is followed by a backslash to guarantee TeX compatibility.
>
>
> **Q3. Please explicitly describe the training pipeline (router via (3)–(4), commander via (5)–(7)), and state whether any joint or end-to-end optimization of  and  is performed. If not, rephrase Eq. (2) or present a bilevel/alternating optimization view.**
>
> **Answer 3:** In the revised version, we have added a detailed description of the training procedure in Section 2.1 (Problem Formulation), explicitly explaining how the router operates through Equations (3)–(4), and how the commander performs task aggregation through Equations (5)–(7). We have also made it clear that the training of the router and the commander is not conducted jointly or in an end-to-end manner.

---

### Review · Reviewer_xkLK · 2025-09-17

**Summary Of Contributions:**

This paper proposes Commander-GPT, a modular multi-agent framework for multimodal sarcasm detection. Instead of relying on a single monolithic model, it decomposes the task into six cognitively grounded subtasks. A central “commander” dynamically routes each input to appropriate specialist agents, collects their outputs, and integrates them to make the final decision. The framework is evaluated on the MMSD and MMSD 2.0 benchmarks across various commander models (from fine-tuned BERT to large models like GPT-4o and Gemini Pro). Experiments show consistent improvements in F1 scores over several prompting-based baselines, with the largest gains observed for smaller and mid-sized models.

**Strengths**

- The paper identifies a well-motivated problem by focusing on the difficulty of multimodal sarcasm detection. The proposed approach shows that decomposing sarcasm understanding into six sub-modules is cognitively plausible and helps explainability, addressing the black box nature of existing LLM-based sarcasm detectors.
- The proposed framework is quite flexible and model agnostic. Experiments show that the CommanderGPT architecture can orchestrate various LLM/MLLM backbones (from small to large), showing robustness and adaptability.
- The authors evaluate across multiple commander sizes and provide ablations, subtask invocation frequency analysis, routing visualizations, and out-of-distribution tests, which strengthen the empirical claims.
- Experiments indicate sizable F1 score improvements, especially in resource-constrained settings, showing that structured reasoning can compensate for smaller model capacity.

**Weaknesses**

- While modularity can bring performance gains, the paper does not report latency or computational overhead of routing and invoking multiple agents, which could be significant in real applications.
- The routing classifier is trained on ~5,000 GPT-4o-labeled samples, but the quality and diversity of these annotations are not analyzed. If these labels are biased or limited, routing could overfit or fail in unseen scenarios.
- Details about how the commander models are trained (especially the encoder-based version) are somewhat sparse.
- While performance gains are reported, there is little qualitative or category-wise error analysis showing where Commander-GPT helps or still fails (eg., subtle sarcasm, implicit sentiment contradiction, etc.).

**Audience:**

Yes

**Audience Explanation:**

The paper focuses on a problem setting of interest to the broader ML community with practical application scenarios, and the proposed approach would provide an interesting case study of a simplistic approach to leveraging multiple models and prompt structure for achieving success in an otherwise difficult task.

**Broader Impact Concerns:**

The work poses no major ethical risks, but since Commander-GPT is trained on social media datasets (e.g. MMSD, MMSD 2.0), it may inherit demographic or cultural biases and show uneven performance across groups. It also relies on large models like GPT-4o and Gemini Pro, which can propagate existing biases. A short Broader Impact Statement acknowledging these risks and outlining mitigation plans would be appropriate.

**Claims And Evidence:**

Yes

**Claims Explanation:**

The core performance claims of the paper are largely supported by clear and convincing evidence: Commander-GPT consistently outperforms prompting-based and monolithic baselines on MMSD and MMSD 2.0, with substantial gains across different commander backbones (from BERT to GPT-4o and Gemini Pro). Ablation studies support the contribution of each subtask module, and an additional evaluation on SemEval 2018 Task 3 provides evidence of generalization. However, some claims, particularly about efficiency, scalability, and the reliability of routing are less substantiated, as the paper does not report computational overhead, analyze the quality of the GPT-4o–distilled routing labels, or evaluate on noisy real-world data. Overall, the experimental evidence is convincing for accuracy improvements, but less so for the broader claims of practicality and robustness.

**Requested Changes:**

Following should be addressed by authors for an accept recommendation:

1. Report inference time, number of model calls per sample, and GPU/CPU memory usage.

2. Show examples where Commander-GPT succeeds while other models fail, and vice versa, to illustrate what the subtask decomposition captures.

3. Describe the GPT-4o distillation process in more detail (prompt design, inter-annotator agreement if any, label quality checks).

4. Clarify Commander-GPT model training details with chosen hyperparameters and training duration.

5. Explain the limitations more explicitly by including discussion of scalability challenges, reliance on external models, and how the framework could be made lighter or faster for real-world deployment.

---

> ### Author Response · Authors · 2025-09-30
>
> Dear Editor, Dear Reviewer,
>
> We have carefully revised the paper in response, with changes highlighted in red, and prepared a detailed point-by-point reply.
>
> Sincerely,
>
>
> **Q1. Report inference time, number of model calls per sample, and GPU/CPU memory usage.**
>
> **Answer 1:** In the revised version, we have added a report of inference time in Section 3.3 (Main Results), including the number of model calls per sample as well as GPU/CPU memory usage. In particular, we discuss the inference latency of Commander-GPT under different configurations, along with its memory consumption (see Fig. 3). These results provide valuable references for evaluating the efficiency and deployment of the framework.
>
>
> **Q2. Show examples where Commander-GPT succeeds while other models fail, and vice versa, to illustrate what the subtask decomposition captures.**
>
> **Answer 2:** We have added Section 3.11 Error Analysis, which presents several case studies. Specifically, we illustrate scenarios where Commander-GPT successfully captures sarcasm while other models fail, as well as cases where Commander-GPT misses sarcasm that other models are able to detect. Figure 8 clearly demonstrates this process.
>
>
> **Q3. Describe the GPT-4o distillation process in more detail (prompt design, inter-annotator agreement if any, label quality checks).**
>
> **Answer 3:** In Section 2.3 (Routing Approach), we have added detailed supplements describing the step-by-step process of GPT-4o distillation, including prompt design and annotator consistency checks. We explicitly explain how we leverage a powerful LLM to generate routing activation decisions and ensure the reliability of label quality. In addition, we provide the methodology for constructing the distilled dataset as well as the procedure for label quality verification.
>
>
> **Q4. Clarify Commander-GPT model training details with chosen hyperparameters and training duration.**
>
> **Answer 4:** In the revised version, we provide detailed hyperparameter settings of Commander-GPT in Section 3.2 (Experimental Settings), including batch size, learning rate, and optimizer (Adam), along with a discussion of training time and hardware configuration. In the experimental section, we explicitly state the rationale behind hyperparameter selection and explain how the model training was optimized to improve performance.
>
>
> **Q5. Explain the limitations more explicitly by including discussion of scalability challenges, reliance on external models, and how the framework could be made lighter or faster for real-world deployment.**
>
> **Answer 5:** In Section 4 (Conclusion), we have added a more detailed discussion of the limitations of the Commander-GPT framework. In particular, we address challenges in scalability, such as the adaptability to larger-scale datasets; the framework’s dependency on external models; and our plans to make the framework more suitable for real-world deployment through further model compression and optimization techniques.

---

### Decision · Action_Editor_w8Gn · 2025-11-03

**Recommendation:** Accept as is

**Audience:**

Yes

**Audience Explanation:**

CommanderGPT improves accuracy of sarcasm detection and can provide a good baseline for future work to build upon. Some of the techniques used to route to specialist agents can be used in the future in other works.

**Claims And Evidence:**

Yes

**Claims Explanation:**

The paper illustrates a multi-agent framework for a multi-modal sarcasm recognition task. The system is carefully engineered to contain agents that perform diverse tasks such as context modeling, sentiment analysis, etc. while a commander agent (either a LLM or a multimodal based encoder) is tasked to provide the final decision given all the information sub-agents captured.

Strengths of the paper include empirical gains over a complete set of baselines (especially after the rebuttal) and ablation studies showing the impact of every agent on the full pipeline. Weaknesses include limited technical novelty and dependence on large proprietary LLMs and specialist modules; robustness to weaker/swapped agents was not evaluated.

Addition after the rebuttal were helpful in improving clarity especially around the hyperparameter settings and how the training/distillation is performed.